# Effect of muscle fatigue of the thoracic erector spinae on neuromuscular control when performing the upper extremity functional tasks in people with adolescent idiopathic scoliosis

**Ray Y. H. Chan**[1☺], **Aiden C. F. Ma**[1☺], **Tammy S. K. Cheung**[1☺], **Jenny C. L. Chan**[1☺], **Ruby W. Y. Kwok**[1☺], **Allan C. L. Fu**[2‡], **Sharon M. H. Tsang**[1☺‡*]

1 Department of Rehabilitation Science, The Hong Kong Polytechnic University, Hung Hom, Hong Kong,
2 School of Health Sciences, University of Sydney, Sydney, Australia

☺ These authors contributed equally to this work.
‡ ACLF and SMHT also contributed equally to this work.
* Sharon.Tsang@polyu.edu.hk

**Data Availability Statement:** All relevant data are within the paper.

## Abstract

Adolescent idiopathic scoliosis (AIS) disrupts spinal alignment and increases the intrinsic demand for active stabilization to maintain postural stability. Understanding the paraspinal muscle fatigability and its effects on spinal alignment and kinematics informs the importance of paraspinal muscle endurance for postural stability. This study aims to investigate the effects of fatigue of thoracic erector spinae on the spinal muscle activity and spinal kinematics in individuals with scoliosis. Spinal muscle activity, posture and mobility measured by electromyography and surface tomography were compared between 15 participants with scoliosis and 15 age- and gender-matched healthy controls during unilateral shoulder flexion and abduction with and without holding a 2-kg weight and performed before and after a fatigue task (prone isometric chest raise). No between-groups difference was found for the spinal extensor endurance. Erector spinae activity at the convex side of AIS group was significantly higher than that at their concave side and than that of healthy controls during shoulder elevations, regardless of the fatigue status. Significant decreases in translational and rotational mobility were found at convex side of AIS group during weighted abduction tasks after fatigue. In contrast, a significant increase in rotational mobility was demonstrated at convex side of AIS participants during weighted flexion tasks after fatigue. Our results revealed a comparable level of spinal extensor endurance between individuals with or without AIS. The increase in muscle activation post-fatigue provides no additional active postural stability but may increase the risk of back pain over the convex side in individuals with scoliosis. Findings highlight imbalances in muscles and the potential implications in optimising neuromuscular activation and endurance capacity in the rehabilitation for AIS patients. Future research is needed to investigate if endurance training of the convex-sided back extensors could optimize the impaired neuromuscular control in the AIS patients.

**Funding:** The authors received no specific funding for this work.

**Competing interests:** The authors have declared that no competing interests exist.

## Introduction

Adolescent idiopathic scoliosis (AIS) is a three-dimensional structural deformity of the spine with unknown aetiology. It involves deviations in the coronal and sagittal planes from the physiological curves, usually accompanied by intervertebral rotation [1–3]. AIS is defined by a Cobb Angle greater than 10˚ in the coronal plane and this condition affects approximately 1–4% of adolescents between the ages of 11 and 18, and disproportionately affects young women, with a female-to-male incidence ratio ranging from 1.5:1 to 11:1 [1]. Recent literature has proposed a multifactorial theory that describes AIS as a genetic pathology of the central nervous system, suggesting that asymmetrical muscle recruitment of erector spinae (ES) muscles may be one of the major factors that cause a growing spine to be susceptible to distortions [1, 3].

Previous studies have reported a significant imbalance in spatial activation of the ES between the convex and concave sides of spinal scoliotic curves [3]. Some studies have further demonstrated increased ES activity on the convex side compared with the concave side in AIS subjects with right thoracic curvature, and the activity level of the corresponding side in normal subjects under isometric back extension and extrinsic perturbations [4, 5]. Although asymmetry in muscle activity could not be verified as the cause or result of scoliosis, extant literature asserts that muscle asymmetry of ES in patients with AIS promotes the [3, 6]. Thus, Park et al. [7] suggested strengthening the comparatively weaker ES muscles over the concave side to balance the ES activities on both sides and perhaps help in managing scoliosis. Nonetheless, such a therapeutic exercise approach has not been substantiated by the histological evidence in patients with AIS.

Histology studies of the ES in AIS patients have revealed an increase in the muscle volume and relative composition of Type I muscle fibres as well as a decrease in the Type II fibre ratio at the convex side of the scoliotic curve [5, 8]. Type I muscle fibres are responsible primarily for postural stability and endurance activities, whereas Type II fibres are responsible primarily for high-speed and forceful movements of short duration [9]. Adaptive changes in muscle fibre composition suggest an intrinsic, asymmetrical demand for endurance (an increase in tonic demand on the convex side) that could contribute to the reported asymmetry in spinal kinematics and muscle activation during tasks performed by AIS patients [5]. The opposite pattern in ES muscle composition has been found over the concave side (i.e., a decrease in Type I fibres and an increase in Type II fibres), along with an increase in muscle tone and stiffness [6, 10]. Zapata et al. [11] further explained that increased fibrosis, fatty infiltration, and Type II fibres may contribute to this phenomenon. A motor pattern characterized by stiffness and high muscle tone points towards treatments that help relieve muscle tension over strengthening, and hence it contradicts what the aforementioned electromyographic findings suggest. Due to the structural deviations in spinal alignment in patients with AIS, the ES muscles are identified as adopting a role in maintaining both static and dynamic postural stability [3]. The differences among theories in treating AIS patients in terms of optimising the paraspinal muscle status raise concerns over whether the strengthening of the convex-side ES would be beneficial to all AIS patients. Thus, an understanding of the spinal alignment and kinematics after fatigue of the ES muscles may help inform the role of ES endurance in maintaining better postural stability during upper limb loading tasks.

Performing voluntary arm movement is an established method to challenge equilibrium and postural stability of the spine [12]. The role of paraspinal muscles in adapting to intrinsic perturbation is substantiated by the activation of paraspinal muscle after initial deltoid onset in arm elevation movements as observed by Lee et al. [13]. Deficits in paraspinal muscle strength and endurance due to muscle composition changes in patients with AIS potentially compromise their ability to maintain postural stability, which may be reflected in spinal kinematics.

This study 1) compared the endurance of paraspinal muscles and 2) investigated the effects of fatiguing the thoracic erector spinae (TES) (using simulating endurance challenges) on the kinematics and muscle recruitment of the thoracic spine between individuals with and without AIS. We hypothesized that individuals with scoliosis would have 1) a significantly lower level of endurance in their paraspinal muscles, 2) a significantly higher muscle recruitment of thoracic ES on the convex side than on the concave side of their scoliotic curves, and 3) a significantly greater variability in the movement of the thoracic spine when executing weighted shoulder elevation tasks than when doing non-weighted tasks post-fatigue, compared with individuals without scoliotic curves.

## Materials and methods

### Participants

Fifteen participants with AIS (the AIS group) and fifteen age- and gender-matched healthy controls (the HC group) were recruited through a patient self-help group and the local community between April and October 2021 (Table 1). Ethical approval of this study was obtained from the Institutional Review Board of The Hong Kong Polytechnic University, and the procedures were conducted according to the Declarations of Helsinki. All participants signed the written informed consent documents, and those in the AIS group also completed the adapted Chinese version of the Scoliosis Research Society-22 (SRS-22) questionnaire [14], before the data collection conducted at The Spine Research Laboratory of The Hong Kong Polytechnic University.

Participants recruited in the AIS group were adolescents aged 12–25 years with a Cobb angle greater than 10˚. In contrast, none of the healthy controls demonstrated any significant scoliotic curvature and all had been symptom-free in all their spinal regions during the previous 12 months. Individuals were excluded if they had any other known orthopaedic, neurological, or systemic disorders, any previous surgery or trauma to the brain or spine, or any painful trunk motion [16].

**Instrumentation and measurements.** *Surface tomography for spinal alignment and posture*. The DIERS formetric 4D system (DIERS International Gmbh, Schlangenbad, Germany) was used to analyse spinal alignment and posture. Ten reflective markers were placed on the landmarks of each participant's body, enabling the system to establish the correspondence between the surface and the underlying skeleton. For the cervical region, markers were placed at the C7 spinous process and the left and right acromion processes. For the thoracic and

**Table 1. Descriptive statistics for the participants (mean ± SD).**

| Variable | AIS (n = 15) | Healthy (n = 15) | p-value |
|---|---|---|---|
| Age (years) | 18.93 ± 4.11 | 19.93 ± 2.28 | 0.910 |
| Gender (female:male) | 10:5 | 8:7 | 0.456 |
| Height (m) | 1.64 ± 0.81 | 1.68 ± 0.08 | 0.257 |
| Weight (kg) | 52.80 ± 10.73 | 57.40 ± 8.63 | 0.702 |
| BMI (kg/m$^2$) | 19.58 ± 2.87 | 20.32 ± 1.63 | 0.857 |
| Exercise regularly* | 8 | 5 | 0.269 |
| Time until exhaustion (s) (Prone isometric chest raise) | 174.13±71.71 | 198.73±63.94 | 0.330 |

BMI = Body Mass Index

*Exercise regularly is defined as participating in moderate-intensity physical activities for more than 150 minutes/week or in vigorous-intensity physical activities for more than 75 minutes/week [15].

lumbopelvic regions, markers were placed at the T3, T6, T9, T12, and L3 spinous processes, and the left and right posterior superior iliac spines (PSISs) (Fig 1). Each participant was positioned 2 m from a formetric 4D projection and camera unit, which projected stripes of light onto the surface of the participant's back [17]. Using rasterstereography, the system recorded the surface of the participant's back, non-invasively and radiation-free. The camera unit captured images of the subject's back at a rate of 1 frame/s during the upper limb tasks. The combination of images formed a real-time three-dimensional representation of the shape of the spine in motion, thus enabling evaluation of spinal kinematics during task execution [18].

*Electromyography and kinematics of spine and shoulders.* Electromyographic (EMG) activity was recorded by surface electromyography using the MyoMuscle System 9000 (Noraxon Inc., USA) at a sampling frequency of 1024 Hz. Six pairs of muscles––the bilateral cervical erector spinae (CES), the thoracic erector spinae (TES) at T4 (TES4) and T9 (TES9) levels and the lumbar erector spinae (LES) were studied. Disposable disc-shaped (10-mm⌀) bipolar Ag–AgCl surface EMG electrodes were placed over the corresponding muscles, in accordance with the SENIAM recommendations, with an inter-electrode distance of 20 mm [19], as follows:

- CES: 2 cm lateral to the C4 spinous process bilaterally, over the muscle belly

- TES4 and TES9: 4 cm lateral to the T4 (TES4) and T9 (TES9) spinous processes bilaterally, over the muscle belly

- LES (longissimus): two-finger width lateral to the L1 spinous process

The skin area for EMG electrode placement was lightly abraded with sandpaper and cleaned with alcohol swabs to reduce the skin's electrical impedance to below 5 kΩ. Maximum voluntary contraction (MVC) of each muscle group was obtained for comparisons of the level of effort required of the muscles for the corresponding functional tasks.

In addition to the EMG measurements, synchronized measurement of the kinematic trajectory of the shoulders during the tasks was conducted by an electromagnetic tracking system

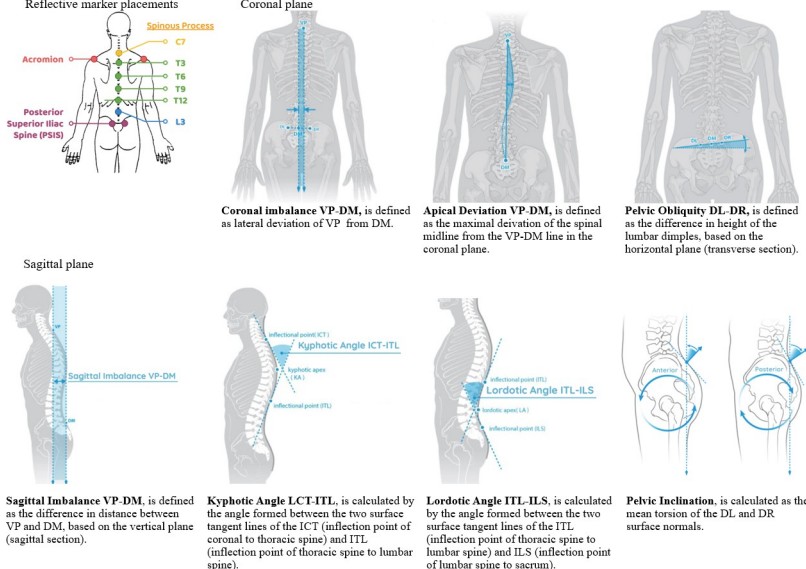

**Fig 1. Reflective marker placement; Kinematic variables measured by the DIERS formetric 4D system.**
Abbreviations: DL: sacral dimple left; DR, sacral dimple right; DM: middle point between DL and DR; ICT: cervicothoracic transition point; ILS: lumbosacral transition point; ITL: thoracolumbar transition point; KA: kyphotic angle; LA: lordotic angle; SP: sacral point; VP: vertebral prominence.

(MyoMotion System, Noraxon Inc., USA) at a sampling frequency of 120 Hz, with wearable sensors secured over the spinous process of T1 and the bilateral upper arms.

## Experimental procedures

Fig 2 presents an overview of the experimental procedures.

**Standardized functional tasks of the shoulders.**   Functional tasks (shoulder flexion and abduction) were employed to induce internal perturbations for active postural muscle adjustment [12]. Before the test, the participants positioned their arms by their sides. They were asked to perform maximum shoulder flexion and abduction under standardized instructions (Fig 3), after practising 3–5 times before the data collection to ensure task familiarization. No verbal feedback was provided to correct the movement during the assessment, because the study sought to investigate the natural movement occurring at the thoracic spine during the functional tasks. The pace was governed by a metronome set at 1 beat/s, with each cycle taking 4 s to complete (2 s for elevation and 2 s for depression) to minimize the confounding effect of the movement speed [20]. A rest interval of 30 s was provided between all tasks, and the level of discomfort during the tasks was closely monitored. The tasks were terminated immediately if the participants experienced any discomfort or pain greater than a mild degree (i.e., Numeric Pain Rating Scale > 3/10). The participants then repeated the functional tasks while holding a 1.5-kg weight in each hand [21].

The sequence of the functional tasks performed was:

1. Left shoulder flexion x 2 times

2. Right shoulder flexion x 2 times

3. Left shoulder abduction x 2 times

4. Right shoulder abduction x 2 times

Immediately after the fatigue procedure, the participants were asked to perform the same sequence of functional tasks again in both the weighted and non-weighted conditions, for reassessment.

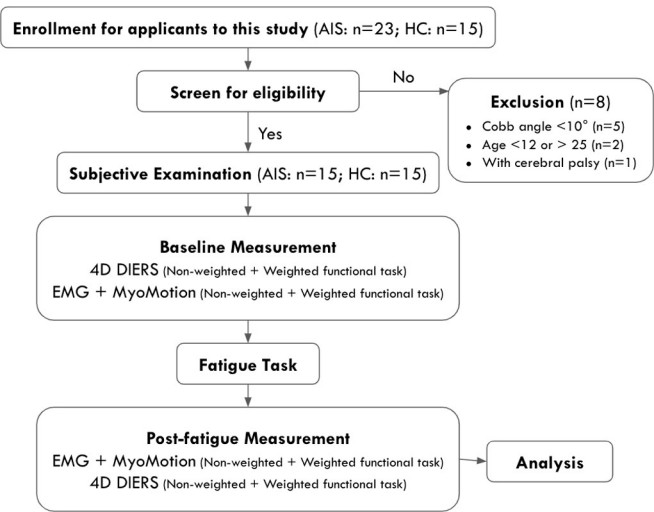

**Fig 2. Flow chart of experimental procedures.**

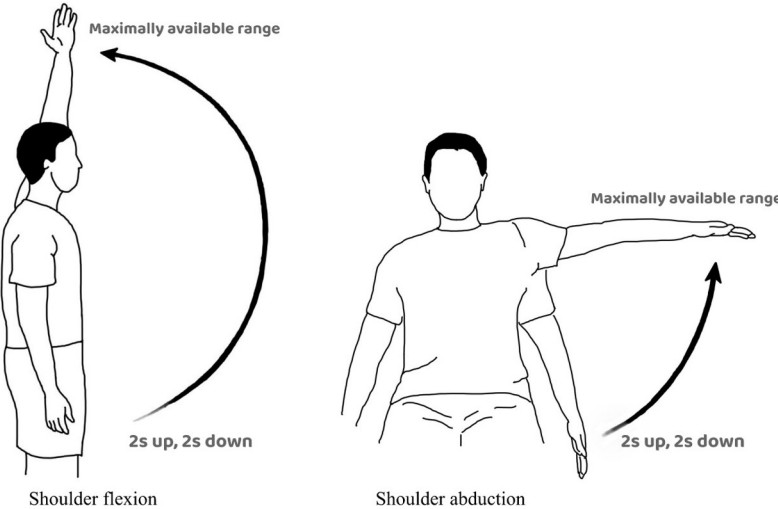

**Fig 3. Standardized functional tasks–shoulder flexion and shoulder abduction.**

**Fatigue procedure for the thoracic erector spinae (TES).** The prone isometric chest raise test, a common clinical muscle endurance test, was used to induce fatigue of the erector spinae muscles and then to evaluate their endurance, using time. Two pillows were put underneath the participant's abdomen to facilitate the person's lift-off of the upper trunk and prevent hyperlordosis and/or muscle effort of the lumbar spine. The participants were instructed to place their arms by their sides and keep their sternum off the table. They had to raise their upper trunk up by 30˚, flex their neck, and hold that position for as long as they could tolerate it while maintaining normal breathing [21]. The intraclass correlation coefficient of the test was 0.95, suggesting that it was a reliable test [22]. The median frequency (MF) of the TES EMG activity obtained by spectrum analysis was assessed to verify the condition of muscle fatigue.

## Data processing and statistical analyses

Statistical analyses were performed using IBM SPSS 25.0 statistical software (SPSS Inc., Chicago, USA).

**Endurance and fatigability of the thoracic erector spinae (TES) muscles.** The durations of the prone isometric chest raise test and the values of MF were obtained from the spectrum analysis of EMG signals from the TES4, TES9, and LES muscles at five different timeframes (0%, 25%, 50%, 75%, and 100% of the overall duration). Comparisons between the healthy controls and the AIS individuals, and the convex side and the concave side of individuals with AIS, were drawn using the independent *t*-test, with the level of significance set at 0.05.

**Kinematics and EMG data.** The mobility of the participants' spines, as measured by spinal tomography during the execution of the functional tasks, was reflected by analysing kinematic variables in two planes. Sagittal imbalance, kyphotic angle, lordotic angle, and pelvic inclination were the variables in the sagittal plane, whereas coronal imbalance, pelvic obliquity, and apical deviation were measured in the coronal plane.

To analyse the electromyographs, raw EMG signals were filtered (using band-pass filtering at 10–500 Hz), the full wave was rectified, and smoothing was processed with the root mean square (RMS) over a moving average of 10 Hz. An EMG of each muscle was obtained at the 90˚ arm elevation for each movement cycle verified by the shoulder kinematics. The EMG

activation patterns of the eight pairs of selected muscles were analysed in terms of their level of effort, as expressed in percentage of MVC (%MVC). Changes in the listed kinematics and the level of effort of the corresponding muscles during the functional tasks were compared between groups and for time-and-group interactions, using mixed model ANOVA analysis.

## Results

No significant differences in age, gender, height, weight, body mass index (BMI), or exercise regularity were found between the two groups (Table 1). All AIS participants except one had a C-curve, with most of them located in the thoracolumbar region. The average Cobb angle was 30.07˚, and the average SRS-22 score was 3.89 (Table 2), suggesting that the participants had moderate scoliosis [23].

### Electromyographic activity during the prone isometric chest raise test

No significant between-group differences were found in the average times until exhaustion ($p$ = 0.330) (Table 1). Both the AIS group and the HC group demonstrated a decline in MF in the thoracic and lumbar ES muscles (Table 3), verifying the presence of muscular fatigue induced by the prone isometric chest raise test. Although the lumbar ES muscles were more prone to fatigue than the thoracic ES muscles in the HC group, no significant differences were found in the AIS group.

The muscle activation pattern during the fatigue task was also investigated to reveal the relative contributions among the regional trunk extensors (Fig 4). A comparable pattern was observed between the two groups, in which the LES demonstrated a decreasing trend in muscle activation. However, the LES muscles were still activated most throughout the fatigue task. Meanwhile, the contributions of the TES4 and TES9 muscles increased with time, as reflected by their rising %MVC values.

### Electromyographic activity during functional tasks

Because no significant differences were found in any of the eight pairs of muscles in their %MVC between the left and right sides for the HC group, the mean EMG values for the paired sides of a muscle (HC$_{mean}$) were compared with the muscle activations of the convex (AIS$_{convex}$) and concave sides (AIS$_{concave}$) in the AIS group. Generally, the AIS participants had a consistently higher %MVC value over the convex side than they did over the concave side in the TES4 and TES9 muscles (Table 4).

Fig 5 illustrates the %MVC values for the HC participants and the convex and concave sides of the AIS participants during the weighted and non-weighted functional tasks at pre-

**Table 2. Characteristics of the AIS group.**

| Cobb angle (˚) | 30.07 ± 12.05 | | |
|---|---|---|---|
| Type of curve / Level of apex | Main thoracic (n = 1) | T7 | |
| | Thoracolumbar (n = 12) | T9 (3) | L1 (4) |
| | | T12 (3) | L2 (2) |
| | Lumbar (n = 1) | L3 | |
| | Double major (n = 1) | T8 & L2 | |
| SRS-22 SUM | 85.56 ± 9.42 | | |
| SRS-22 Mean | 3.89 ± 0.41 | | |

SRS-22 = the adapted Chinese version of the Scoliosis Research Society-22 questionnaire

**Table 3. Medium frequency slope values of TES4, TES9 and LES during the fatigue task.**

| Group and Side | Muscle | Mean | 95% CI—Upper bound | p-Value |
|---|---|---|---|---|
| **AIS Convex** | TES 4 | -0.072 | -0.023 | 0.381 |
| | TES 9 | -0.090 | -0.055 | 0.637 |
| | LES | -0.140 | -0.099 | |
| **AIS Concave** | TES 4 | -0.067 | -0.017 | 0.387 |
| | TES 9 | -0.071 | -0.043 | 0.450 |
| | LES | -0.128 | -0.083 | |
| **Healthy Control Left** | TES 4 | -0.0922 | -0.044 | **0.002**[*] |
| | TES 9 | -0.127 | -0.080 | **0.031**[*] |
| | LES | -0.231 | -0.175 | |
| **Healthy Control Right** | TES 4 | -0.090 | -0.036 | **0.000**[*] |
| | TES 9 | -0.135 | -0.079 | **0.017**[*] |
| | LES | -0.245 | -0.192 | |

[*]Indicates significant difference in medium frequency slope values ($p < 0.05$) between TES4/TES9 and LES

and post-fatigue. A significant time effect was observed in the CES and TES4 muscles in the non-weighted conditions, whereas a significant time effect was found in all muscles except the LES in the weighted conditions.

Significant time-and-group interactions were found only in the TES9 muscles in the weighted conditions. Post-hoc analysis with Bonferroni correction revealed significant interactions between the $AIS_{convex}$ and $HC_{mean}$ values in the TES9 ($p = 0.044$) muscle.

## Spinal kinematics during functional tasks

No significant differences were found between the left-side and right-side movement for all seven kinematic variables, in both the AIS and HC groups (Table 5). Hence, the average range of the spinal movement for the two sides in the HC group was compared with that of the convex and concave sides in the AIS group.

Figs 6 and 7 show the range of spinal movement in the sagittal and coronal planes respectively. A significant time effect was observed in the kyphotic angle and the apical deviation during non-weighted conditions, as well as in coronal imbalance and pelvic obliquity during weighted conditions. Nevertheless, there was no significant time-and-group interaction in any of the kinematic variables.

Because scoliosis occurs in the coronal plane, further analyses were conducted separately for the coronal-plane variables during weighted flexion and abduction. The results revealed a significant time-and-group interaction between the AIS group's convex-side task and the mean for the control group in coronal imbalance in the weighted abduction task ($p = 0.046$), and in apical deviation in both the weighted flexion ($p = 0.016$) and abduction tasks ($p = 0.047$) (Fig 8).

## Discussion

### Endurance test and time until exhaustion

Our findings of a negative slope in the median frequency and an increase in the %MVC in the TES4 and TES9 muscles in both the AIS and HC groups verify the establishment of fatigue in the thoracic ES [24, 25]. The healthy controls presented a significantly greater decline in MF over the LES muscles compared with those for the TES4 and TES9 during the fatigue task.

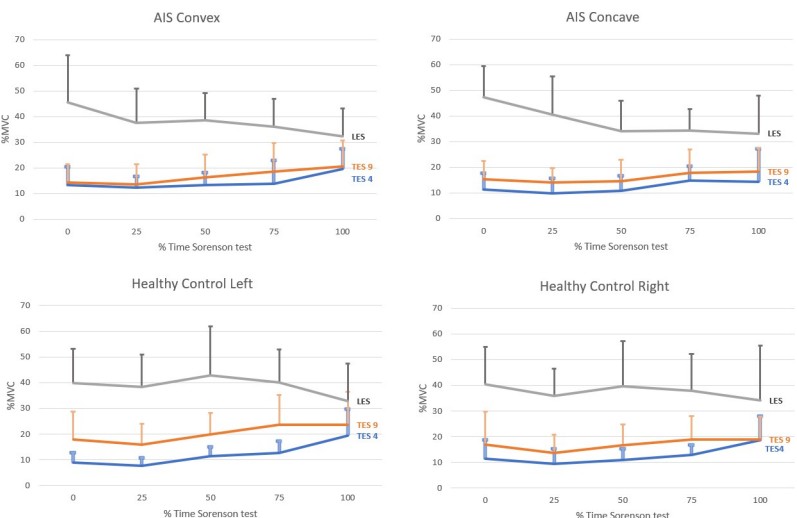

**Fig 4. Muscle activation (expressed in percentage of maximal voluntary contraction [%MVC]) of TES4, TES9 and LES during the fatigue task (mean + SD).**

Meanwhile, no significant between-group differences in the drop of MF were found with the TES4, TES9, and LES values in the AIS group. This implies a comparable degree of fatigue among the three sets of muscles, thus suggesting a lack of selective regional control during the fatigue task.

Although our first hypothesis was rejected by the insignificant between-groups differences in the average time until exhaustion, we noted that the AIS group exhibited a wider range of performance (AIS: 90–300 s vs. HC: 127–300 s) with dichotomized exercise habits (regular exercisers vs. sedentary individuals). The percentage of participants engaged in regular exercise was substantially higher in the AIS group (AIS: 53.3% vs HC: 33.3%). Thus, our finding may suggest that in general AIS individuals would benefit from being physically active, as was reflected by the AIS participants' comparable level of trunk muscle endurance with that of the healthy controls.

### General spinal muscle activity

Our study found a significantly higher level of activity of the TES4 and TES9 muscles on the convex side than on the concave side of the scoliotic curve, irrespective of the task or fatigue effect, and this muscle activation pattern is generally consistent with previous findings in other

**Table 4. Comparison of muscle activation (expressed in percentage of the maximal voluntary contraction [%]) between convexity and concavity in AIS group and left and right sides in healthy controls.**

|  | AIS (mean ± SD) | | | Healthy Control (mean ± SD) | | |
|---|---|---|---|---|---|---|
|  | Convex | Concave | p-Value | Left | Right | p-Value |
| **CES** | 22.46 ± 13.42 | 24.16 ± 15.97 | 0.374 | 17.82 ± 11.74 | 18.24 ± 10.09 | 0.769 |
| **TES4** | 25.18 ± 16.80 | 20.95 ± 14.24 | **0.037**[*] | 20.11 ± 12.08 | 17.83 ± 10.49 | 0.121 |
| **TES9** | 24.85 ± 12.34 | 20.90 ± 10.70 | **0.009**[*] | 15.88 ± 8.16 | 16.35 ±8.79 | 0.670 |
| **LES** | 9.75 ± 5.62 | 8.50 ± 5.64 | 0.088 | 5.66 ± 3.93 | 5.75 ± 3.54 | 0.854 |

Cervical erector spinae (CES); Thoracic erector spinae at T4 (TES 4) and T9 (TES 9); Lumbar erector spinae (LES)

[*]Indicates significant difference in %MVC (p<0.05) between two sides, independent from the functional tasks and experimental (fatigue) procedures

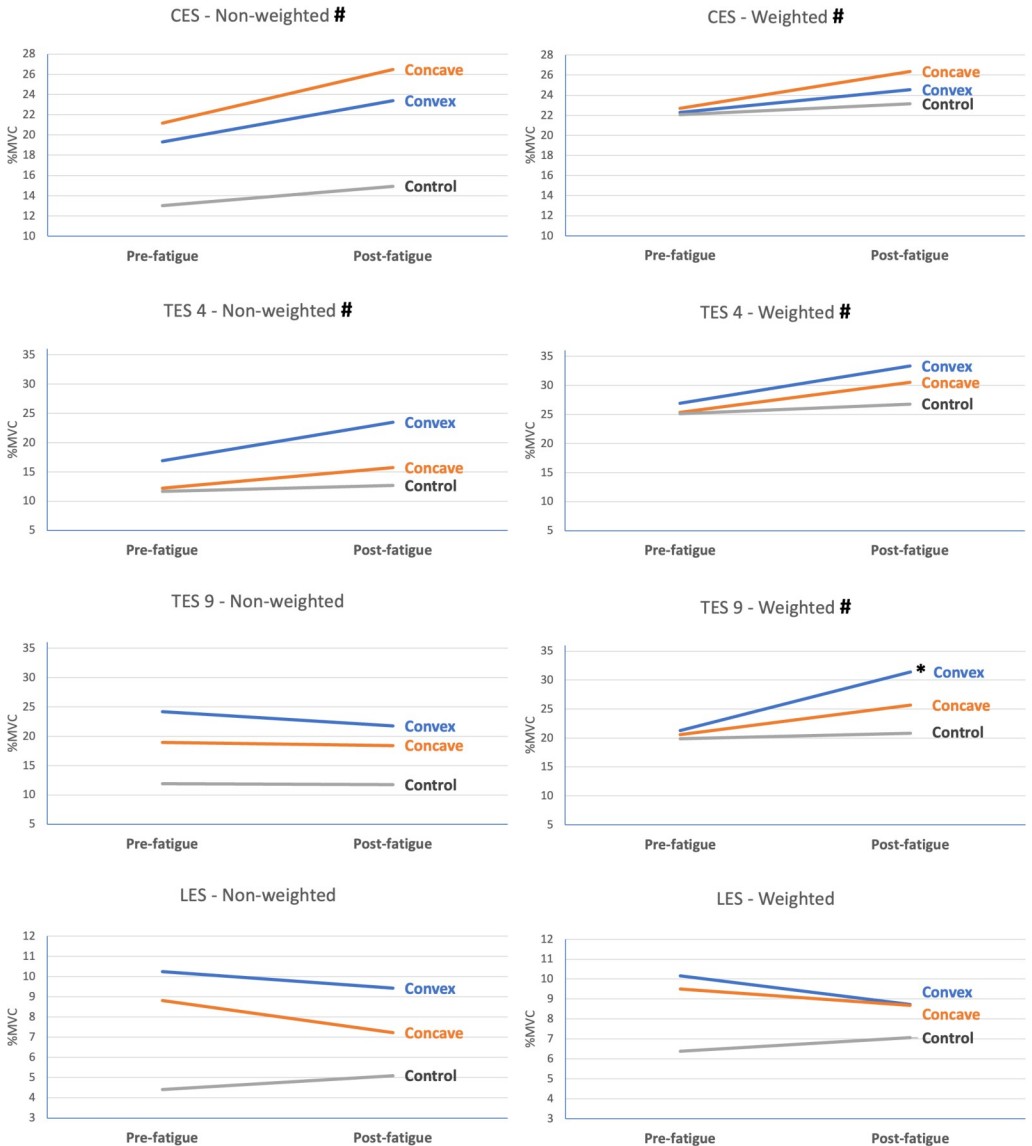

**Fig 5. Muscle activation (expressed in percentage of maximal voluntary contraction [%MVC]) during weighted and non-weighted functional tasks at pre- and post-fatigue.**

tasks [4, 5]. Park et al. [7] reported a greater activity level of the TES at T12 but not at T7, over the convex side in AIS individuals with thoracolumbar curves, during a 10-s trunk extension in individuals lying prone. Although both Park et al. [7] and our study measured subjects with thoracolumbar curves, our findings revealed a significantly higher TES4$_{convex}$ activity and insignificant differences in LES muscle activation in the AIS group. This may be attributable to the different nature of the tasks in which an upper limb elevation task was adopted in our study.

## Muscle activity of the TES9 muscles

Within the TES muscles, the spinalis thoracis lies in a groove centrally located between the vertebral column and the spinous process and extends from T2 to T8 with segmental attachments,

**Table 5. Comparison of spinal kinematics between convex-sided with concave-sided shoulder elevation in AIS group and between left and right shoulder elevation in healthy controls.**

|  | AIS (mean ± SD) | | | Healthy Control (mean ± SD) | | |
|---|---|---|---|---|---|---|
| Range | Convex | Concave | p-Value | Left | Right | p-Value |
| **Sagittal Imbalance (˚)** | 1.34 ± 0.90 | 1.26 ± 0.79 | 0.490 | 1.04 ± 0.56 | 1.14 ± 0.73 | 0.210 |
| **Kyphotic Angle (˚)** | 3.23 ± 2.31 | 2.93 ± 1.61 | 0.237 | 3.50 ± 2.15 | 4.01 ± 2.19 | 0.071 |
| **Lordotic Angle (˚)** | 6.51 ± 7.03 | 6.01 ± 6.73 | 0.632 | 6.23 ± 6.60 | 7.16 ± 6.69 | 0.281 |
| **Pelvic Inclination (˚)** | 2.13 ± 1.55 | 2.37 ± 1.79 | 0.273 | 2.84 ± 3.67 | 3.57 ± 4.18 | 0.150 |
| **Coronal Imbalance (˚)** | 1.81 ± 1.54 | 1.67 ± 1.25 | 0.437 | 1.15 ± 0.56 | 1.30 ± 1.06 | 0.176 |
| **Pelvic Obliquity (˚)** | 1.30 ± 0.90 | 1.37 ± 0.85 | 0.557 | 1.15 ± 0.66 | 1.29 ± 0.88 | 0.175 |
| **Apical Deviation (mm)** | 5.50 ± 2.89 | 4.95 ± 2.65 | 0.124 | 3.74 ± 2.31 | 4.19 ± 2.26 | 0.127 |

Convex: convex-side arm elevation; Concave: concave-side arm elevation; Left: left arm elevation; Right: right arm elevation

whereas the longissimus thoracis attaches to the transverse processes of each thoracic and lumbar vertebra and to the 2nd through 12th ribs [26]. Because of this characteristic, the TES may have a dual function of serving as the primary mover of trunk extension and side-flexion and secondarily of assisting in global spinal stabilization. In addition, the TES muscles' role in spinal stabilization for individuals with AIS is postulated to be even more vital because of the three-dimensional spinal distortion. Coronal deviation may induce a negative impact associated with the excessive elongation of the $TES_{convex}$, which further compromises spinal stability [8]. As our findings show, the AIS participants adopted a compensatory strategy by recruiting the $TES_{convex}$ at a higher level in an attempt to accommodate the functional demand.

A significant time-and-group interaction was found between the AIS individuals' $TES9_{convex}$ activity and the healthy individuals' TES9 activity––the segment that was close to the apex of the curves for our AIS participants. A significantly greater increase in the AIS groups' $TES9_{convex}$ activity at the post-fatigue trial suggests that it has higher susceptibility to detrimental impacts related to muscle fatigue. Histological evidence of the increment in muscle volume and relative composition of Type I muscle fibres at the convex side [5, 8] has been found predominantly at the apex of the AIS curve. Hence, it is logical to explain the significantly higher post-fatigue $TES9_{convex}$ activity in the AIS group compared with the activity in the corresponding muscles of the control group. This finding also implies that a greater effort of the $TES_{convex}$ is required in AIS individuals to achieve postural stability, due to their impaired neuromuscular efficiency [27].

Higher activity of the $TES9_{convex}$ muscles in AIS individuals may be an adaptive compensatory mechanism to address an acute postural stability demand, but unfortunately, it may become hazardous to muscle health if this is prolonged. In AIS individuals, the %MVC of $TES9_{convex}$ exceeded 30% in weighted upper limb functional tasks post-fatigue, which could impair the vascularity to the muscles. Studies have revealed that muscle contractions at an MVC level between 20% and 40% can significantly deprive and even occlude the vascular circulation [28]. Miyake et al. [29] also reported significantly prolonged post-fatigue recovery of muscles at the convex side compared with the concave side in patients with degenerative lumbar scoliosis, which signified a compromised oxygen-retaining capacity in their $TES_{convex}$. These findings together suggest a higher risk of ischemia in $TES9_{convex}$ muscles, and may explain the higher prevalence of back pain experienced by AIS individuals over their convexity side [30]. The 1.5-kg weight used in our task is considered to be essentially low relative to the load demands in our daily activities. Therefore, when the trunk extensors of AIS individuals are chronically fatigued, such as after carrying a heavy school bag with insufficient rest, the

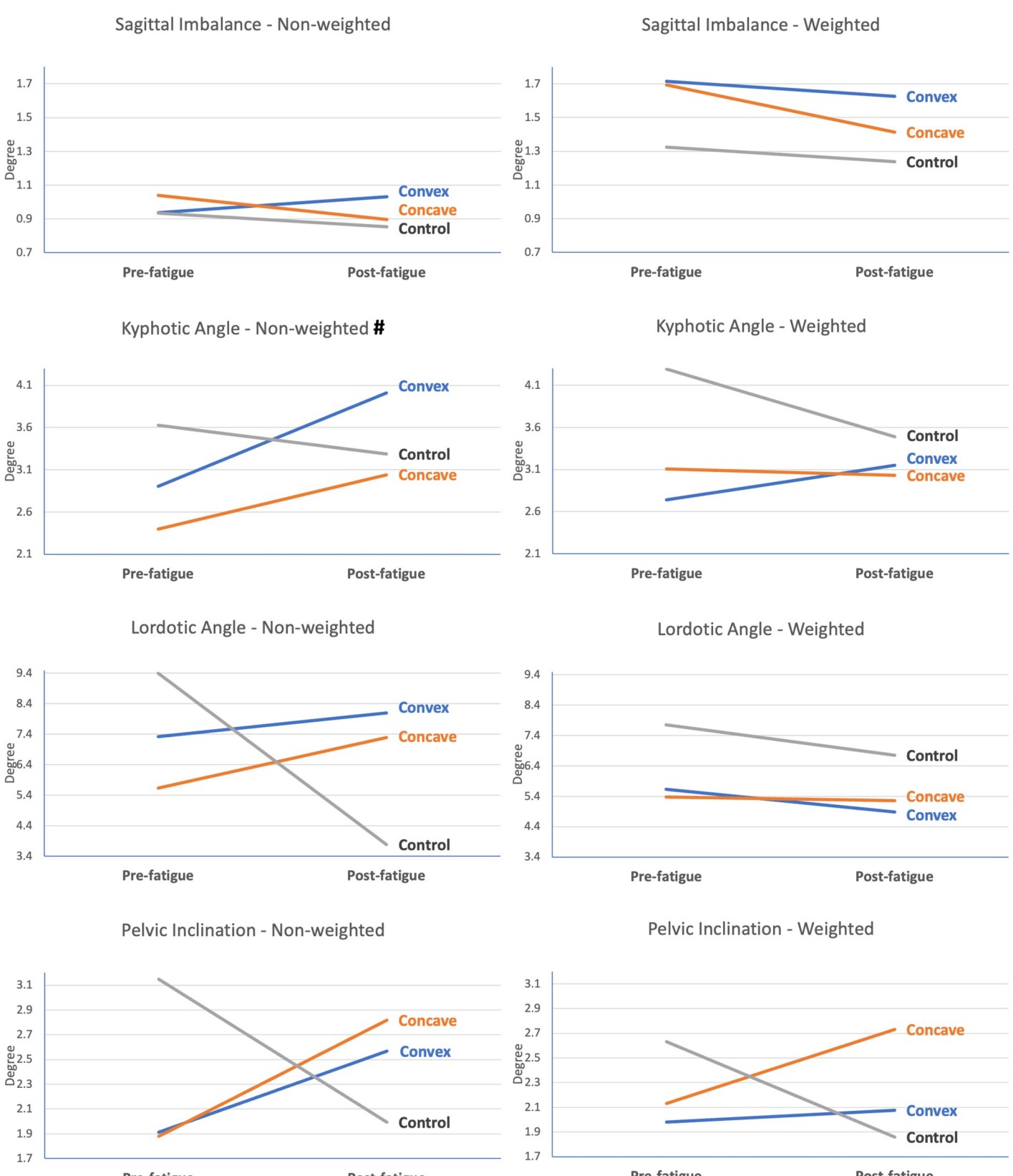

**Fig 6. Sagittal-plane spinal kinematics during weighted and non-weighted functional tasks at pre- and post-fatigue.** # represents significant time effect between pre- and post- fatigue measurement (p<0.05).

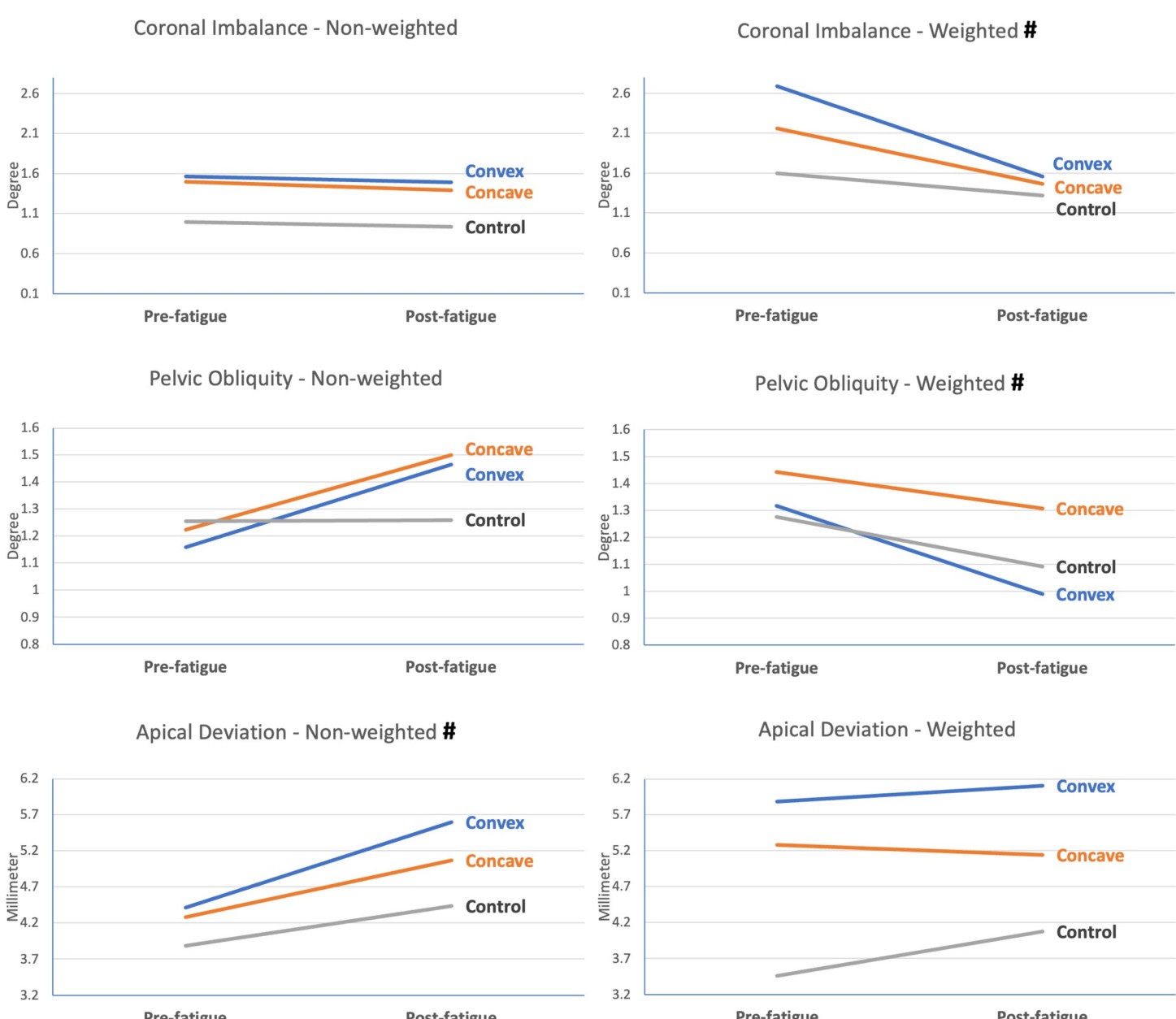

**Fig 7. Coronal-plane spinal kinematics during weighted and non-weighted functional tasks at pre- and post-fatigue.** # represents significant time effect between pre- and post- fatigue measurement (p<0.05); * represents significant time-and-group interaction between the indicated side and control group (p<0.05).

mechanical stress induced in AIS individuals may be multiplied and may contribute to the development or perpetuation of back pain.

## Spinal kinematics

Higher EMG activity of the $TES9_{convex}$ in the AIS individuals compared with that in the corresponding muscles in the healthy individuals after the fatigue task implies greater engagement of the active subsystem in postural stability, concurring with the reduced kinematic values in the coronal plane during weighted abduction task. For global postural stability, a significantly

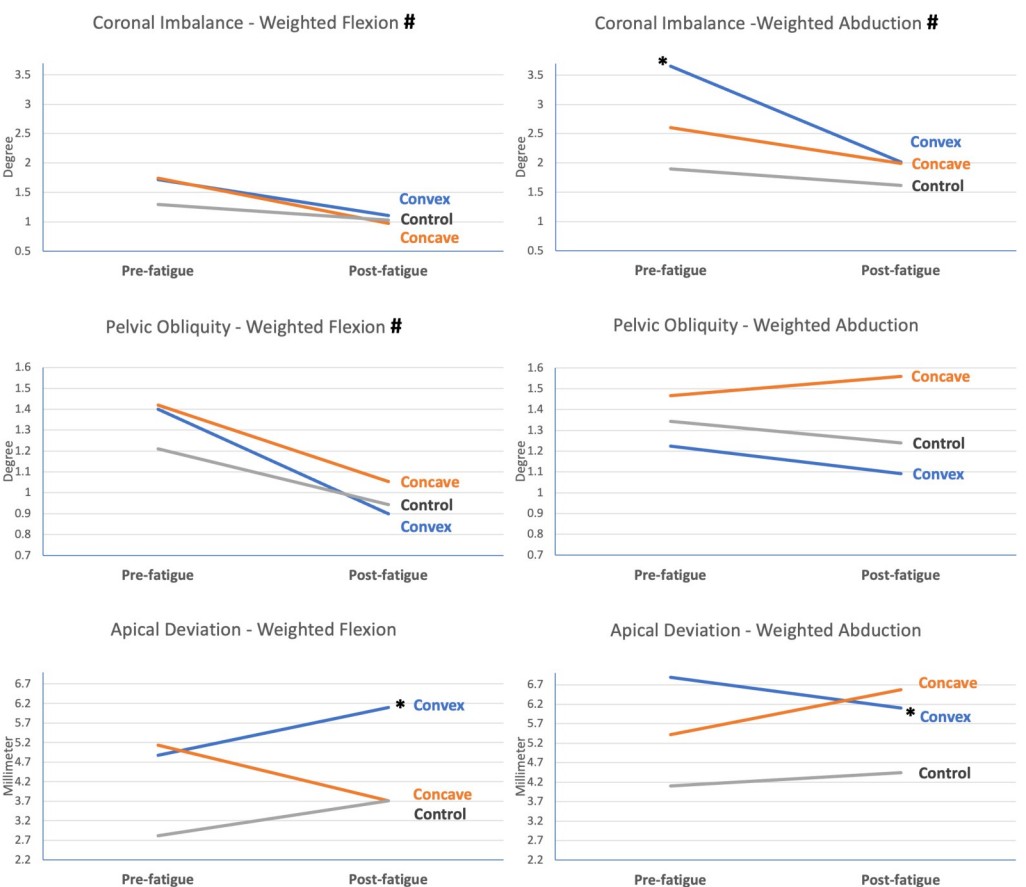

**Fig 8. Coronal-plane spinal kinematics during weighted flexion and weighted abduction tasks of the shoulder joint.**
# represents significant time effect between pre- and post- fatigue measurement (p<0.05); * represents significant time-and-group interaction between the indicated side and control group (p<0.05).

greater decrease in the range of coronal imbalance during convex-side weighted arm abduction compared with healthy individuals may imply an increased postural stability post-fatigue [31]. The same findings occurred in segmental stability, as reflected by the decreased range of apical deviation.

Nevertheless, the active spinal stabilization provided by the TES$_{convex}$ muscles may become inefficient in weighted arm flexion under the post-fatigue circumstance. For convex-side weighted flexion tasks, the range of apical deviation increased significantly more than that of the healthy individuals did after fatigue, thus implying poorer spinal stability. As mentioned, the TES muscles have a primary role in trunk extension and assist secondarily in global spinal stabilization. Because the capacity to achieve the maximal force production had proved to be decreased after muscle fatigue [32], we postulate that on the convex side during the weighted flexion task, the TES$_{convex}$ muscles mainly worked as a spinal extensor to counteract the torque induced by a long lever arm from the upper limb with weight. Therefore, despite their increase in activity, the contribution of the ES muscles in global spinal stabilization may be greatly reduced, as reflected by our findings of a significantly greater increase in apical deviation. Because most functional activities are multi-planar, the increase in active spinal stability during shoulder abduction in the coronal plane may not translate into functional activities.

## Conclusions

Although this study found no significant difference in trunk extensor endurance between the AIS and healthy individuals, the AIS individuals' TES9 convex muscles was found to be more adversely affected by the fatigue of trunk extensors than were the corresponding muscles in the healthy individuals. The increase in activation of these muscles post-fatigue provides no additional active postural stability and may increase the risk of back pain over the convex side in AIS individuals. Due to the differential demand between convex and concave sides of para-spinal muscles revealed in AIS individuals, further research is required to examine whether back endurance training, especially over the convex side, may be beneficial to the AIS populations. Effects of scoliosis on spinal kinematics in the sagittal plane, center of pressure, and bilateral upper limb tasks should also be investigated to better inform the clinical evaluation and long-term management of AIS.

## Acknowledgments

The authors would like to acknowledge the assistance of Hong Kong Scoliosis Awareness Group for recruitment of the participants and the participants who have joint this study.

## Author Contributions

**Conceptualization:** Ray Y. H. Chan, Aiden C. F. Ma, Tammy S. K. Cheung, Jenny C. L. Chan, Ruby W. Y. Kwok, Sharon M. H. Tsang.

**Data curation:** Ray Y. H. Chan, Aiden C. F. Ma, Tammy S. K. Cheung, Jenny C. L. Chan, Ruby W. Y. Kwok, Sharon M. H. Tsang.

**Formal analysis:** Ray Y. H. Chan, Aiden C. F. Ma, Tammy S. K. Cheung, Jenny C. L. Chan, Ruby W. Y. Kwok, Allan C. L. Fu, Sharon M. H. Tsang.

**Investigation:** Ray Y. H. Chan, Aiden C. F. Ma, Tammy S. K. Cheung, Jenny C. L. Chan, Ruby W. Y. Kwok, Sharon M. H. Tsang.

**Methodology:** Ray Y. H. Chan, Aiden C. F. Ma, Tammy S. K. Cheung, Jenny C. L. Chan, Ruby W. Y. Kwok, Allan C. L. Fu, Sharon M. H. Tsang.

**Project administration:** Sharon M. H. Tsang.

**Resources:** Sharon M. H. Tsang.

**Supervision:** Sharon M. H. Tsang.

**Validation:** Sharon M. H. Tsang.

**Writing – original draft:** Ray Y. H. Chan, Aiden C. F. Ma, Tammy S. K. Cheung, Jenny C. L. Chan, Ruby W. Y. Kwok, Allan C. L. Fu, Sharon M. H. Tsang.

**Writing – review & editing:** Ray Y. H. Chan, Aiden C. F. Ma, Tammy S. K. Cheung, Jenny C. L. Chan, Ruby W. Y. Kwok, Allan C. L. Fu, Sharon M. H. Tsang.

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
