## [Decision Letter · Decision Letter 0]

18 Nov 2022

PONE-D-22-23148Effect of Muscle Fatigue of the Thoracic Erector Spinae on Neuromuscular Control When Performing the Upper Extremity Functional Tasks in People with Adolescent Idiopathic ScoliosisPLOS ONE

Dear Dr. Tsang,

Thank you for submitting your manuscript to PLOS ONE. After careful consideration, we feel that it has merit but does not fully meet PLOS ONE’s publication criteria as it currently stands. Therefore, we invite you to submit a revised version of the manuscript that addresses the points raised during the review process.

Dear Authors,an expert in the field reviewed your manuscript and found some minor issues you should address in the revision process.

We look forward to receiving your revised manuscript.

Kind regards,

Emiliano Cè

Academic Editor

PLOS ONE

Journal Requirements:

Reviewers' comments:

Reviewer's Responses to Questions

**Comments to the Author**

1. Is the manuscript technically sound, and do the data support the conclusions?

Reviewer #1: Yes

2. Has the statistical analysis been performed appropriately and rigorously? 

Reviewer #1: Yes

3. Have the authors made all data underlying the findings in their manuscript fully available?

Reviewer #1: Yes

4. Is the manuscript presented in an intelligible fashion and written in standard English?

Reviewer #1: Yes

5. Review Comments to the Author

Reviewer #1: Dear Editor,

the manuscript entitled "Effect of Muscle Fatigue of the Thoracic Erector Spinae on Neuromuscular Control When Performing the Upper Extremity Functional Tasks in People with Adolescent Idiopathic Scoliosis" compared the neuromuscular and kinematic response of paravertebral muscles and of the spine during fatiguing protocol in a group of adolescent affected by idiopathic scoliosis and in a group of healhty age-matched subject.

The rationale of the study is clear and the reading fluent. The results support the conclusion tread by the authors.

I have just few minor comments to improve the manuscript.

1. At the beginning of the Introduction the authors wrote "Being the most common form of scoliosis, AIS is defined by a Cobb Angle greater than 10° in the coronal plane." I'd suggest to remove "Being the most common form of scoliosis" as it is not the cause for the sentence that follows it.

2. "Some studies have further demonstrated increased ES activity on the convex side compared with the concave side in AIS subjects, and in normal subjects under isometric back extension and extrinsic perturbations (4, 5)". I find this sentence unclear. It seems that normal subjects have a convex and a concave side but it is not so. Moreover, if isometric back extension is a symmetrical movement (backward), how can be identified a convex and a concave side in normal subject under isometric back extension?

3. I suggest to move "The opposite pattern in ES muscle composition has been found over the concave side (i.e., a decrease in Type I fibres and an increase in Type II fibres), along with an increase in muscle tone and stiffness (6, 10)" immediately after "Histology studies of the ES in AIS patients have revealed an increase in the muscle volume and relative composition of Type I muscle fibres as well as a decrease in the Type II fibre ratio at the convex side of the scoliotic curve (5, 8)."

4. In Participants paragraph some XXXX chars are present.

5. Please, correct SEIAM with SENIAM In "Electromyography and kinematics of spine and shoulders" paragraph.

6. I kindly recommend the authors to add a picture describing the functional task (shoulder flexion and abduction).

7. Measure unit in Table 3 and 4 are lacking.

8. From figure 4 to figure 6 the quality of the picture is poor. I kindly ask the authors to increase it and to add both the measure unit and the standard deviation.

6. PLOS authors have the option to publish the peer review history of their article (what does this mean?). If published, this will include your full peer review and any attached files.

Reviewer #1: No

---

## [Author Response · Author response to Decision Letter 0]

23 Dec 2022

PONE-D-22-23148R1

Effect of Muscle Fatigue of the Thoracic Erector Spinae on Neuromuscular Control When Performing the Upper Extremity Functional Tasks in People with Adolescent Idiopathic Scoliosis

PLOS ONE

Response to Review Comments 

The rationale of the study is clear and the reading fluent. The results support the conclusion tread by the authors. I have just few minor comments to improve the manuscript.

Reviewer’s comments: 

1. At the beginning of the Introduction the authors wrote "Being the most common form of scoliosis, AIS is defined by a Cobb Angle greater than 10° in the coronal plane." I'd suggest to remove "Being the most common form of scoliosis" as it is not the cause for the sentence that follows it.

Response: Thank you for your recommendation. This sentence has been revised as, “AIS is defined by a Cobb Angle greater than 10° in the coronal plane and this condition affects approximately 1-4% of adolescents between the ages of 11 and 18, and disproportionately affects young women, with a female-to-male incidence ratio ranging from 1.5:1 to 11:1 (1).” (line 32-33). 

2. "Some studies have further demonstrated increased ES activity on the convex side compared with the concave side in AIS subjects, and in normal subjects under isometric back extension and extrinsic perturbations (4, 5)". I find this sentence unclear. It seems that normal subjects have a convex and a concave side but it is not so. Moreover, if isometric back extension is a symmetrical movement (backward), how can be identified a convex and a concave side in normal subject under isometric back extension?

Response: Thank you for your comment. It was the value of the electromyography activity of the corresponding side of the spinal muscle in normal subjects being used to compare with those found on the convex side and concave side of the same paraspinal muscles in the AIS subjects whom have the right-sided curvature. Therefore, this sentence has been revised to improve its clarity as, “Some studies have further demonstrated increased ES activity on the convex side compared with the concave side in AIS subjects with right thoracic curvature, and the activity level of the corresponding side in normal subjects under isometric back extension and extrinsic perturbations (4, 5) (line 43-44).

3. I suggest to move "The opposite pattern in ES muscle composition has been found over the concave side (i.e., a decrease in Type I fibres and an increase in Type II fibres), along with an increase in muscle tone and stiffness (6, 10)" immediately after "Histology studies of the ES in AIS patients have revealed an increase in the muscle volume and relative composition of Type I muscle fibres as well as a decrease in the Type II fibre ratio at the convex side of the scoliotic curve (5, 8)."

Response: Thank you for your suggestion. We understand the suggestion of having this specific sentence moved immediately after the sentence that reports the findings of the histology studies. We have tried to move it as suggested and edit the flow of the elaboration of the relevant impacts in AIS, however, it was finally decided to keep this sentence as in the original version i.e., elaborates the functions of the respective muscle type before contrasting the differences in the erector spinae muscles of those with AIS, for better flow of our illustration of the discrepancy between the healthy controls and AIS population (line 53-65). Thank you for your kind understanding. 

4. In Participants paragraph some XXXX charts are present.

Response: The identity of the ethics review board and research laboratory were covered for the review purpose and the information has been included in this revision version, “Ethical approval was obtained from the Institutional Review Board of The Hong Kong Polytechnic University.” (line 101-102), and “All participants signed the written informed consent documents, and those in the AIS group also completed the adapted Chinese version of the Scoliosis Research Society-22 (SRS-22) questionnaire (14), before the data collection conducted at The Spine Research Laboratory of The Hong Kong Polytechnic University.” (line 106-107).

5. Please, correct SEIAM with SENIAM In "Electromyography and kinematics of spine and shoulders" paragraph.

Response: The acronym of Surface ElectroMyoGraphy for the Non-Invasive assessment of Muscles has been corrected as SENIAM (line 149).

6. I kindly recommend the authors to add a picture describing the functional task (shoulder flexion and abduction). 

Response: A graphic illustration of the functional tasks executed by the participants has been included in the revision (Figure 3).

7. Measure unit in Table 3 and 4 are lacking. 

Response: For Table 3, the results presented the slope (gradient of change over time) of the median frequency of the respective muscle contraction during the Sorensen test, in which this parameter does not have the unit itself. For Table 4, the muscle activation was expressed in terms of the percentage of maximal voluntary contraction of the corresponding muscles, hence, the unit (%) has been added to the table. Thank you.

8. From figure 4 to figure 6 the quality of the picture is poor. I kindly ask the authors to increase it and to add both the measure unit and the standard deviation. 

Response: These three figures have been prepared with high resolution to improve the quality (they are now Figure 5 to 7 as the numbering of the figures has been updated for the inclusion of the additional figure illustrating the functional tasks executed by the participants [Figure 3]). We totally agreed that it is important to include standard deviations of the data similar to the way we presented our result in Figure 4 (originally labelled as Figure 3 which reports the percentage of maximal voluntary contraction of the spinal muscles during the fatigue task), we have in fact included them in all our figures in our preparation version. In response to this specific comment, we have tried again to solve this problem by adjusting the scale of the y-axis, selecting only the plus or minus value of the standard deviations for respective line, and changing the format of the graphic presentation. Unfortunately, we failed to improve the clarity due to the compact and spread of the results (a sample of figure with standard deviation is presented below). Therefore, to balance the pros and cons, we decided to leave it as the original version. Thank you for your kind understanding. 

(A sample figure with inclusion of standard deviation can be found in the attached word file saved as "Responses to reviewer comments").

End of response letter, thank you for reviewing our revised manuscript.

---

## [Decision Letter · Decision Letter 1]

13 Jan 2023

Effect of Muscle Fatigue of the Thoracic Erector Spinae on Neuromuscular Control When Performing the Upper Extremity Functional Tasks in People with Adolescent Idiopathic Scoliosis

PONE-D-22-23148R1

Dear Dr. Tsang,

We’re pleased to inform you that your manuscript has been judged scientifically suitable for publication and will be formally accepted for publication once it meets all outstanding technical requirements. As you can see, the reviewer suggests to reduce to three decimal the data reported in Table 3. Please, consider to do this correction while proofreading the pre-final version of the manuscript.

Kind regards,

Emiliano Cè

Academic Editor

PLOS ONE

Additional Editor Comments (optional):

Reviewers' comments:

Reviewer's Responses to Questions

**Comments to the Author**

1. If the authors have adequately addressed your comments raised in a previous round of review and you feel that this manuscript is now acceptable for publication, you may indicate that here to bypass the “Comments to the Author” section, enter your conflict of interest statement in the “Confidential to Editor” section, and submit your "Accept" recommendation.

Reviewer #1: All comments have been addressed

2. Is the manuscript technically sound, and do the data support the conclusions?

Reviewer #1: Yes

3. Has the statistical analysis been performed appropriately and rigorously? 

Reviewer #1: Yes

4. Have the authors made all data underlying the findings in their manuscript fully available?

Reviewer #1: Yes

5. Is the manuscript presented in an intelligible fashion and written in standard English?

Reviewer #1: Yes

6. Review Comments to the Author

Reviewer #1: Dear Authors,

all my comments received a feedback and almost all of them have been addressed.

I kindly recommend to reduce the number of decimals in table 3. In my opinion 3 decimals are enough.

7. PLOS authors have the option to publish the peer review history of their article (what does this mean?). If published, this will include your full peer review and any attached files.

Reviewer #1: No

---

## [Editor Report · Acceptance letter]

19 Jan 2023

PONE-D-22-23148R1 

Effect of Muscle Fatigue of the Thoracic Erector Spinae on Neuromuscular Control When Performing the Upper Extremity Functional Tasks in People with Adolescent Idiopathic Scoliosis 

Dear Dr. Tsang:

I'm pleased to inform you that your manuscript has been deemed suitable for publication in PLOS ONE. Congratulations! Your manuscript is now with our production department. 

Kind regards, 

on behalf of

Professor Emiliano Cè 

Academic Editor

PLOS ONE